# LKB1 regulates JNK-dependent stress signaling and apoptotic dependency of *KRAS*-mutant lung cancers

Chendi Li[1,2], Mohammed Usman Syed [1], Anahita Nimbalkar[1], Yi Shen[1], Melissa D. Vieira[1], Cameron Fraser [3,4], Zintis Inde [3,4], Xingping Qin [3,4], Jian Ouyang[1,5], Johannes Kreuzer[1,2], Sarah E. Clark[1], Grace Kelley[1], Emily M. Hensley[1], Robert Morris[1], Raul Lazaro[6], Brian Belmonte[6], Audris Oh[1], Makeba Walcott[1], Christopher S. Nabel[1,7], Sean Caenepeel[6], Anne Y. Saiki [6], Karen Rex[6], J. Russell Lipford[6], Rebecca S. Heist[1,2], Jessica J. Lin [1,2], Wilhelm Haas[1], Kristopher Sarosiek [3,4,8], Paul E. Hughes[6] & Aaron N. Hata [1,2] ✉

The efficacy of molecularly targeted therapies may be limited by co-occurring mutations within a tumor. Conversely, these alterations may confer collateral vulnerabilities that can be therapeutically leveraged. *KRAS*-mutant lung cancers are distinguished by recurrent loss of the tumor suppressor *STK11*/LKB1. Whether LKB1 modulates cellular responses to therapeutic stress seems unknown. Here we show that in LKB1-deficient *KRAS*-mutant lung cancer cells, inhibition of KRAS or its downstream effector MEK leads to hyperactivation of JNK due to loss of NUAK-mediated PP1B phosphatase activity. JNK-mediated inhibitory phosphorylation of BCL-XL rewires apoptotic dependencies, rendering LKB1-deficient cells vulnerable to MCL-1 inhibition. These results uncover an unknown role for LKB1 in regulating stress signaling and mitochondrial apoptosis independent of its tumor suppressor activity mediated by AMPK and SIK. Additionally, our study reveals a therapy-induced vulnerability in LKB1-deficient *KRAS*-mutant lung cancers that could be exploited as a genotype-informed strategy to improve the efficacy of KRAS-targeted therapies.

Mutations in KRAS, a small GTPase that regulates MAPK/ERK signaling, define the largest genetically-defined subset of non-small cell lung cancer (NSCLC), representing 25–30% of all lung adenocarcinomas[1]. The recent US FDA and European Commission approvals of sotorasib (AMG 510)[2] and adagrasib (MRTX849)[3], small molecule covalent KRAS^{G12C}-selective inhibitors, marked a milestone in the development of targeted therapies for

*KRAS*-mutant cancers. While most NSCLC patients treated with sotorasib experience clinical benefit, only ~40% achieve a partial response[4]. To improve efficacy, drug combination strategies that target mechanisms of adaptive resistance[5–8] or immune evasion are being tested in the clinic.

*KRAS*-mutant lung cancers harbor diverse co-occurring mutations[1], and although not yet fully characterized, emerging evidence indicates

[1]Massachusetts General Hospital Cancer Center, Boston, MA, USA. [2]Department of Medicine, Massachusetts General Hospital and Harvard Medical School, Boston, MA, USA. [3]John B. Little Center for Radiation Sciences, Harvard T.H. Chan School of Public Health, Boston, MA, USA. [4]Department of Environmental Health, Harvard T.H. Chan School of Public Health, Boston, MA, USA. [5]Department of Biochemistry & Molecular Biology and Hollings Cancer Center, Medical University of South Carolina, Charleston, SC, USA. [6]Amgen Research, Amgen Inc., Thousand Oaks, CA, USA. [7]Koch Institute for Cancer Research, Massachusetts Institute of Technology, Cambridge, MA, USA. [8]Lab for Systems Pharmacology, Harvard Program in Therapeutics Science, Harvard Medical School, Boston, MA, USA. ✉e-mail: ahata@mgh.harvard.edu

that some mutations may predict lack of response to different therapies. For instance, *STK11*/LKB1 loss and *KEAP1* mutations may contribute to lack of response to different therapies including anti-PD-(L)1 immune checkpoint inhibitors[9,10] and KRAS[G12C] inhibitors[3,4]. Co-occurring mutations that positively predict clinical response to KRAS[G12C] inhibitors or drug combinations remain undefined. Considering the genetic heterogeneity of *KRAS*-mutant lung cancers, and the multitude of drug combinations entering clinical testing, it is crucial to identify vulnerabilities conferred by specific genomic alterations and develop biomarkers that can predict response to KRAS[G12C] inhibitor combinations that may help guide patient selection.

Preclinical studies have demonstrated that knockdown or suppression of KRAS or downstream signaling in *KRAS*-mutant cell lines often fails to induce apoptosis[11–13]. Suppression of MEK/ERK signaling leads to the accumulation of the pro-apoptotic BCL-2 family protein BIM, which is critical for inducing apoptosis in response to an array of targeted therapies[14,15]. However, induction of BIM by MEK or KRAS[G12C] inhibition alone is often insufficient to induce apoptosis in *KRAS*-mutant cancer cells because BIM is bound and neutralized by pro-survival BCL-2 family proteins such as BCLX-XL or MCL-1. Combining MEK inhibitors with BH3 mimetics, which competitively bind to BCL-XL or MCL-1 and liberate BIM, can induce apoptosis and lead to regression of *KRAS*-mutant tumors[12,16,17]. However, clinically relevant biomarkers that can differentiate specific apoptotic dependencies (MCL-1 versus BCL-XL) and thus stratify patients for treatment with KRAS[G12C] inhibitor + BH3 mimetic combinations are lacking.

While studying the response of *KRAS*-mutant lung cancer models to KRAS[G12C] or MEK inhibitors combined with BH3 mimetics, we unexpectedly observed an association between the presence of *STK11* mutations and dependence on MCL-1. *STK11*, which encodes the protein LKB1 (Liver Kinase B1), is inactivated in ~30% of *KRAS*-mutant lung

cancers[18]. Loss of the tumor suppressor *STK11*/LKB1[19] facilitates tumorigenesis by modulating energy balance[20,21], enhancing metastatic potential[22,23], and enabling immune evasion[9,24]. However, the role of LKB1 in modulating cellular response to therapy is largely unexplored. Here, we demonstrate that LKB1 regulates JNK stress signaling and the apoptotic response of cancer cells independent of its tumor suppressor activity mediated by AMPK[25–27] and SIK[28,29] kinases. In LKB1-deficient *KRAS*-mutant lung cancer cells treated with KRAS or MEK inhibitors, hyperactivation of JNK occurs due to loss of NUAK-mediated PP1B phosphatase activity. Phosphorylation of BCL-XL by JNK causes a reciprocal dependency on MCL-1, rendering LKB1-deficient cells vulnerable to MCL-1 inhibition. Additionally, our study suggests LKB1 loss as a genotypic marker for sensitivity to KRAS[G12C]-selective inhibitors + MCL-1 inhibitors in LKB1-deficient *KRAS*-mutant lung cancer cells.

## Results

### LKB1 loss confers sensitivity to MAPK + MCL-1 inhibition

To investigate the impact of common co-occurring genomic alterations on KRAS[G12C] inhibitor combination strategies targeting distinct pathways, we screened a panel of *KRAS[G12C]*-mutant NSCLC cell lines harboring diverse co-occurring mutations (Fig. S1A) with sotorasib alone or in combination with inhibitors targeting SHP2 (TNO 155), CDK4/6 (abemaciclib), PI3K (GDC-0941), BCL-XL/BCL-2 (navitoclax) or MCL-1 (AMG 176) (Fig. 1A). Consistent with prior studies of KRAS[G12C] inhibitors[2,7,30,31], we observed varying sensitivity to KRAS[G12C] inhibition, which in our limited cell line cohort was independent of the most common co-occurring mutations such as *TP53*, *STK11*/LKB1 and *KEAP1* (Fig. S1B, C; Supplementary Data 1). In general, the overall sensitivity of the cell line panel to the other inhibitors was low, although a minority of cell lines exhibited variable sensitivity to BCL-XL or MCL-1 inhibition (Fig. S1D). This was also independent of co-occurring mutations in our cell line

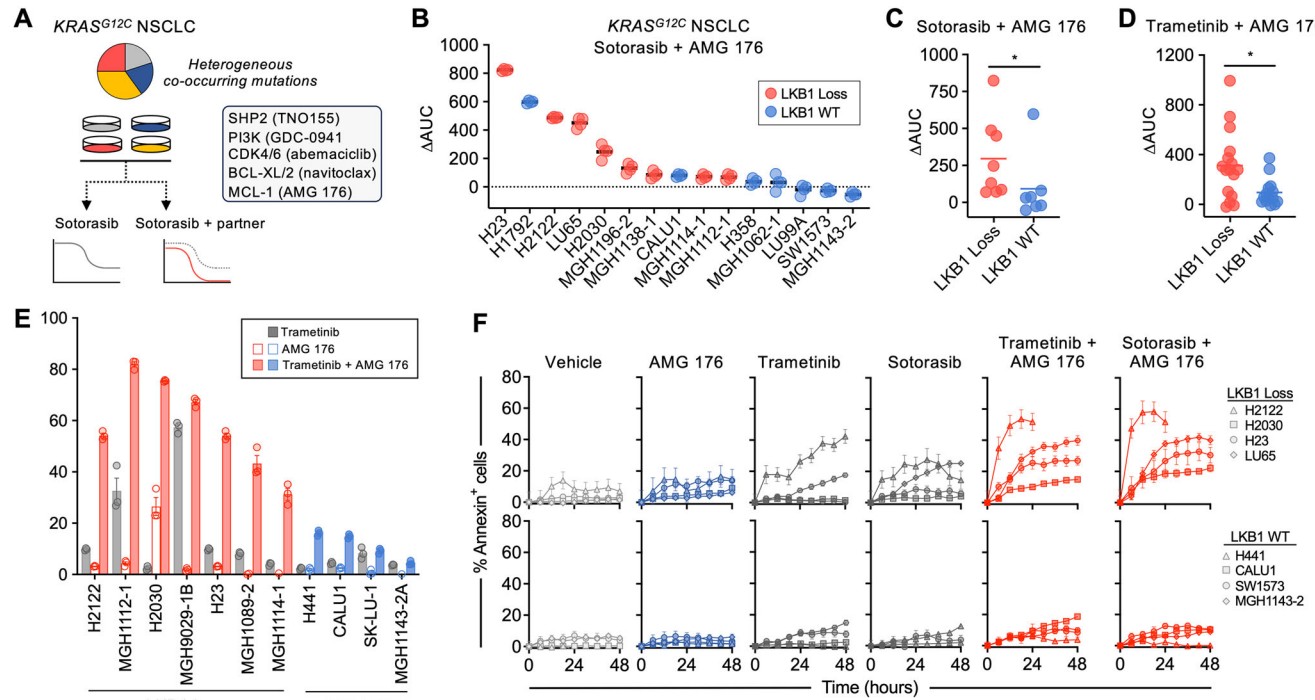

**Fig. 1 | LKB1 loss confers sensitivity to combined MAPK + MCL-1 inhibition in *KRAS*-mutant NSCLC models. A** Schema for testing sotorasib drug combinations. **B** Relative increased efficacy of sotorasib + AMG 176 combination compared to sotorasib alone (ΔAUC−see Fig. S2A for explanation) against *KRAS[G12C]*-mutant NSCLC cell lines. Each dot represents an independent biological replicate, *N* = 4). Comparison of ΔAUC between *KRAS*-mutant NSCLC cell lines stratified according to *LKB1* status. *p = 0.029 (**C**), *p = 0.032 (**D**), unpaired-nonparametric *t* test,

2-sided. *KRAS*-mutant NSCLC cell lines were treated with 0.1 μM of trametinib or 1 μM of sotorasib in combination with 1 μM of AMG 176 for up to 72 h and apoptosis was assessed by annexin positivity by flow cytometry (**E**, data are mean and S.E.M. of *N* = 3 biological replicates) or live-cell imaging (**F**, data are mean and S.E.M. of 3 technical replicates). For annexin positivity, percentage of apoptotic cells in vehicle group was used as a control and normalized to 0. Source data are provided as a Source Data file.

cohort (Fig. S1E). To quantify the efficacy of KRAS[G12C] combinations compared to KRAS[G12C] alone, we calculated the relative change in AUC (e.g., the area between the single agent and combination dose response curves, normalized to the effect of sotorasib alone), referred to hereafter as simply ΔAUC (Fig. S2A). As expected, combining sotorasib with other inhibitors led to greater suppression of cell viability than single-agent sotorasib in most cell lines, although the effect was variable (Fig. S2B). Whereas the presence of co-occurring mutations did not appear to impact sensitivity to combinations targeting SHP2, CDK4/6, or BCL-XL/ BCL-2, cell lines with co-occurring mutations or loss of *STK11*/LKB1 were more sensitive to combinations targeting MCL-1 or PI3K (Fig. 1B, C, Fig. S2B). PI3K inhibition can effect diverse cellular changes in oncogene-addicted cancers, including mTOR-dependent down-regulation of MCL-1 protein levels[32,33], which we confirmed (Fig. S2C). To further investigate the role of MCL-1 in a larger cohort of *KRAS*-mutant NSCLC cell lines that included *KRAS* mutations other than G12C, we tested the MEK inhibitor trametinib in combination with AMG 176 (or the related compound AM-8621[16]). Similarly, we observed greater activity of trametinib + AMG 176 in cell lines with LKB1 loss (Fig. 1D, S2D). We also confirmed these findings with additional MEK (cobimetinib) and KRAS[G12C] (adagrasib) inhibitors (Fig. S2E). The increase in combination activity resulted from synergistic activity between trametinib and AMG 176 (Fig. S2F), resulting in a net cytotoxic effect by the combination (Fig. S2G). LKB1-deficient cell lines with high ΔAUC values exhibited robust apoptosis upon combined inhibition of KRAS/MAPK and MCL-1, while the apoptotic response of LKB1 wild-type (WT) cell lines was minimal (Fig. 1E, F), suggesting that LKB1 may modulate apoptotic dependencies of *KRAS*-mutant lung cancers.

## Modulation of LKB1 expression alters apoptotic response

To determine whether LKB1 plays a causal role in tuning the apoptotic response of *KRAS*-mutant NSCLC cells, we restored LKB1 expression in LKB1-deficient cell lines or deleted LKB1 in WT cell lines (Fig. S3A). We confirmed restoration of functional LKB1 activity by an increase in phosphorylation of the canonical LKB1 kinase substrates AMPK and ACC (Fig. S3B, S5B). Re-expression of LKB1 decreased sensitivity to combined sotorasib or trametinib + MCL-1 inhibition, and conversely, CRISPR-mediated deletion of LKB1 sensitized LKB1 WT cells to sotorasib or trametinib + MCL-1 inhibition (Fig. 2A, B, S3C, D). Restoration or deletion of LKB1 did not alter the response to sotorasib alone (Fig. S3E) or alter cell proliferation rate (Fig. S3F), suggesting that the changes in sensitivity to the drug combination that occur upon gain or loss of LKB1 are mediated primarily by differences in MCL-1-dependent regulation of apoptosis. Consistent with this notion, restoration or deletion of LKB1 decreased or increased the apoptotic cell death to sotorasib or trametinib + AMG 176, respectively (Fig. 2C, D, S3G), with restoration of LKB1 expression converting cytotoxic responses to cytostatic responses (Fig. S3H). To confirm these results in vivo, we established isogenic H2030 EV and LKB1 xenograft tumors in mice. First, we confirmed restoration of LKB1 functional activity in established xenograft tumors by an increase in LKB1 and pAMPK staining compared with LKB1-null H2030 EV controls (Fig. S4A). Additionally, we measured the level of several CRTC2/CREB target genes that are repressed by active LBK1-SIK signaling in tumors[34], confirming decreased gene expression in isogenic LKB1-expressing xenograft tumors compared to the LKB1-null controls (Fig. S4B). Similar to the in vitro results, restoration of LKB1 abolished tumor regression of H2030 xenograft tumors in response to sotorasib or trametinib + AMG 176 (Fig. 2E, S4C). Collectively, these results demonstrate that that loss of LKB1 sensitizes *KRAS*-mutant NSCLC cells to combined MAPK + MCL-1 inhibition both in vitro and in vivo.

## JNK activation is associated with MCL-1 dependence

LKB1 is a master serine/threonine kinase that regulates multiple cellular process including growth[26,35], cell metabolism[20,21], and cell polarity[36–38]. We hypothesized that loss of LKB1 rewires downstream kinase signaling networks to confer dependency on MCL-1, especially upon disruption of oncogenic signaling. Supporting this, expression of a kinase-dead LKB1[K781] (kd) mutant[25] did not rescue LKB1-deficient cells from combined MEK + MCL-1 inhibition (Fig. S5A, B), demonstrating that LKB1 catalytic activity is required for the observed difference in drug sensitivity. To identify differences in kinase signaling in *KRAS*-mutant NSCLC cells with or without LKB1, we performed mass spectrometry-based global phosphoproteome profiling[39] of isogenic H2030 (EV, LKB1 and LKB1-kd) and H358 (KO GFP, KO LKB1) cells before and after treatment with trametinib (Fig. 3A). We quantified 27364 unique phosphosites (Fig. S5C, D), then performed phosphosite signature analysis[40] to identify the kinases that were differentially activated in each of these contexts. Consistent with the known effect of MEK inhibition on cell cycle progression[41], we observed down-regulation of cell cycle associated phospho-signatures including cyclin-dependent kinases, ATM, ATR, Aurora Kinase B, and PLK1 in response to trametinib treatment (Fig. S5E). In the absence of drug treatment, there were few statistically significant differences (and no overlap) in kinase signatures between LKB1 wild-type and deficient cells (Fig. S5F), likely a result of the nutrient-rich cell culture environment. To identify drug-induced differences in kinase activity regulated by LKB1, we looked for kinase phospho-signatures that were enriched in trametinib-treated LKB1-deficient cells relative to their wild-type counterparts (H2030 EV versus LKB1, H358 KO LKB1 versus KO GFP) but not enriched in H2030 EV versus kinase-dead LKB1[K871] cells. While several signatures were enriched in trametinib-treated LKB1-deficient cells for either isogenic pair, only one signature – c-Jun N-terminal kinase1 (JNK1) – satisfied these criteria (Fig. 3B). Specifically, the phosphorylation of well-established substrates of JNK1, such as ATF2, JUN and JUNB, increased to a greater extent in H2030 EV and H358 KO LKB1 cells after trametinib treatment compared to their LKB1 wild-type pairs. Next, we performed proteomic analysis of H2030 and H358 isogenic cells after treatment with trametinib + AMG 176. JNK phospho-signatures rapidly (8 h) increased in H358 LKB1 KO cells compared to control cells, and a subset of JNK substrates showed increase phosphorylation in LKB1-deficient H2030 cells (Fig. S5G, H). These results suggest that LKB1 loss is associated with increased JNK activation upon suppression of oncogenic signaling by trametinib or the trametinib + AMG 176 combination.

To confirm these results, we examined JNK Thr183/Tyr185 phosphorylation in H2030 and H358 isogenic pairs. Combined sotorasib or trametinib + AMG 176 treatment led to a rapid time-dependent increase in JNK phosphorylation in H2030 EV cells (Fig. 3C, S6A) and JNK nuclear translocation (Fig. S6B). JNK activation could be suppressed by knockdown of MKK7, which phosphorylates and activates JNK (Fig. S6C). JNK activation was observed as rapidly as 2 h after drug treatment and preceded apoptotic cell death (Fig. S6D), consistent with a proximal role for JNK activation in the apoptotic response. Re-expression of LKB1 suppressed JNK phosphorylation in H2030 cells, and conversely, deletion of LKB1 in H358 cells led to increased phospho-JNK after drug treatment (Fig. 3C, S6A). We extended these findings by comparing the induction of phospho-JNK across a larger cohort of *KRAS*-mutant NSCLC cells treated with trametinib + AMG 176. Despite an expected degree of heterogeneity between cell lines, LKB1-deficient cell lines overall exhibited greater induction of JNK phosphorylation compared to LKB1 wild-type cell lines with wild-type LKB1, with a significant correlation between pJNK induction and combination sensitivity (Fig. S6E, F). Interestingly, the H1792 cell line, which exhibited the greatest drug sensitivity amongst LKB1 wild-type cells (Fig. 1B), displayed robust induction of pJNK (Fig. S6E). Corroborating the results in H2030 cells, re-expression of LKB1 in H23 cells blunted the induction of phospho-JNK in response to trametinib + AMG 176 (Fig. S6G). These data suggest that LKB1 suppresses JNK-dependent stress signaling that occurs upon inhibition of oncogenic signaling.

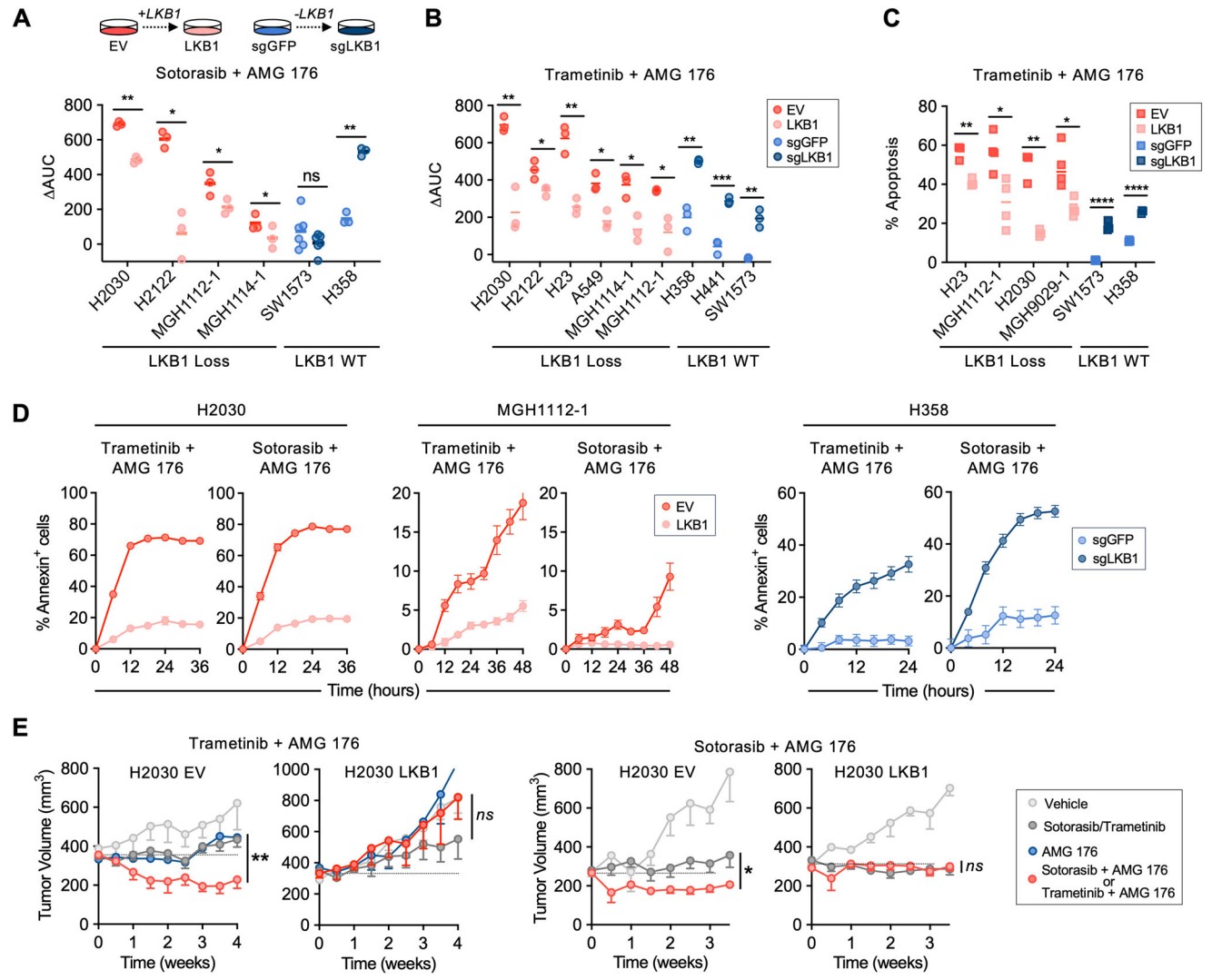

**Fig. 2 | Modulating LKB1 status alters sensitivity to MAPK + MCL-1 inhibition in vitro and in vivo. A, B** Comparison of relative ΔAUC for isogenic LKB1-proficient and deficient *KRAS*-mutant cell line pairs (EV−empty vector, LKB1−LKB1 expression vector, sgGFP or LKB1−CRISPR KO of GFP or LKB1). Each dot represents an independent biological replicate (*N* = 3-6). For Sotorasib, H2030: **p = 0.002, H2122: *p = 0.012, MGH1112-1: *p = 0.014, MGH1114-1: *p = 0.037, H358: **p = 0.007. For Trametinib, H2030: **p = 0.003, H2122: *p = 0.026, H23: **p = 0.002, A549: *p = 0.007, MGH1114-1: *p = 0.011, MGH1112-1: *p = 0.015, H358: **p = 0.0014, H441: ***p = 0.0005, SW1573: **p = 0.002. Paired-parametric *t* test, 2-sided. Apoptotic response of isogenic *KRAS*-mutant NSCLC cell lines after treatment with 0.1 μM trametinib or 1 μM sotorasib in combination with 1 μM of AMG 176 (annexin

positivity assessed by flow cytometry (**C**) or live-cell imaging (**D**). **C** Each dot represents an independent biological replicate, *N* = 3-5, H23: **p = 0.003, MGH1112: *p = 0.015, H2030: **p = 0.0017, MGH9019-2: *p = 0.011, SW1573: ****p = 0.00001, H358: ****p = 0.000015, unpaired-nonparametric *t* test, two-sided. **D** data are mean and S.E.M. of 3 technical replicates. **E** Subcutaneous xenograft tumors were established from H2030 EV and H2030 LKB1 cell lines, and mice were treated with vehicle, sotorasib (30 mg/kg daily), trametinib (3 mg/kg daily), AMG 176 (50 mg/kg daily) or combination. Data shown are mean and S.E.M of *N* = 5-6 mice per arm, statistical difference between single agent and combination arms was determined using mixed 2-way ANOVA effects model, *p = 0.01, **p = 0.0084. Source data are provided as a Source Data file.

As JNKs modulate cell proliferation, differentiation and survival in response a number of different environmental and cellular stressors[42], we examined whether hyperactivation of JNK signaling in LKB1-deficient cells is specific to MAPK inhibition or reflects a more general role for regulation of JNK by LKB1. Upon exposure of H2030 EV or LKB1 cells to UV light, a well-established inducer of JNK signaling[43,44], we observed an increase in phospho-JNK in H2030 EV cells that peaked within 60 min (Fig. S6H). Re-expression of LKB1 reduced UV-induced phospho-JNK in H2030 LKB1 cells, indicating that LKB1 may play a general role in suppressing JNK stress signaling in response to a variety of stimuli. To determine whether JNK activation underlies the increased sensitivity of LKB1-deficient *KRAS*-mutant cancer cells to combined MAPK + MCL-1 inhibition, we used siRNA to simultaneously knock down both JNK1 and 2 isoforms in H2030 cells (Fig. S6I) and assessed the response to combined sotorasib or trametinib + AMG 176.

While JNK1/2 knockdown had little effect on sensitivity to trametinib alone, JNK1/2 depleted cells exhibited decreased sensitivity and apoptotic response to both drug combinations, phenocopying the effect of LKB1 re-expression (Fig. 3D, E, S6J). We extended these findings to three additional LKB1-deficient cell lines (H23, H2122, LU65), in which depletion of JNK1/2 reduced the apoptotic response to trametinib + AMG 176 (Fig. S6K). Collectively, these results suggest that hyper-activation of JNK signaling in the absence of LKB1 increases the MCL-1 dependence of LKB1-deficient *KRAS*-mutant NSCLC cells and sensitizes them to combined KRAS^G12C or MEK + MCL-1 inhibition.

### LKB1 suppresses JNK activation via NUAK1/2 and PP1B
LKB1 exerts its effects via phosphorylation and activation of multiple members of the AMP-activated protein kinase (AMPK) family. For instance, LKB1 plays a central role in energy homeostasis by sensing

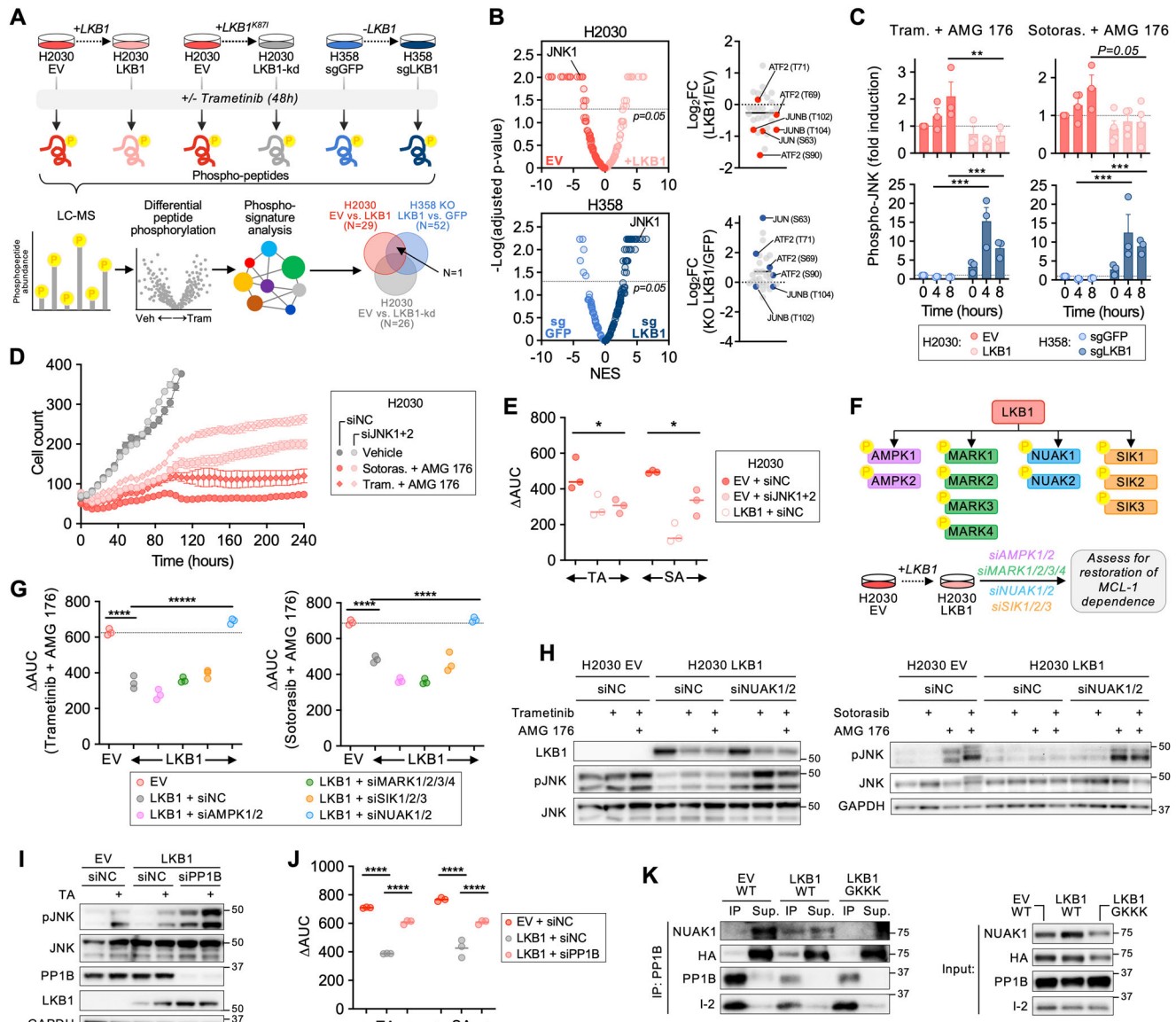

**Fig. 3 | JNK activation in LKB1-deficient cells underlies dependency on MCL-1.**
**A** Phosphoproteomic analysis of isogenic *KRAS*-mutant NSCLC cell lines treated with 0.1 μM of trametinib for 48 h. **B** Left: Differential enrichment of phospho-peptide signatures in trametinib-treated isogenic cell line pairs. NES – normalized enrichment scores. Right, individual phospho-sites of JNK1 downstream substrates are annotated. **C** Change in phospho-JNK in response to MAPK + MCL-1 inhibition in isogenic H2030 and H358 cells (data are mean and S.E.M., each dot represents an independent biological replicate, $N = 3$-4, H2030: **$p = 0.0031$, H358: ***$p = 0.0008$, ***$p = 0.0009$, paired-parametric $t$ test, 2-sided). **D** Change in cell number of H2030 EV cells with siJNK1 + 2 or negative control (siNC) after treatment with 0.1 μM trametinib or 1 μM sotorasib in combination with 1 μM AMG 176 quantified by live-cell imaging. Data are mean and S.E.M. of 3 technical replicates. **E** H2030 EV or LKB1 cells were transfected with siRNAs targeting JNK1 and 2 or siNC and then treated with sotorasib (S) or trametinib (T) ± AMG 176 (A) and viability was determined after 3 days. Each dot represents an independent biological replicate ($N = 3$,

TA: *$p = 0.017$, SA: *$p = 0.028$, unpaired-parametric $t$ test, two-sided). **F** Schematic of siRNA knockdown of LKB1 effectors in H2030 LKB1 cells. **G** H2030 EV or LKB1 cells transfected with corresponding siRNAs were treated with sotorasib or trametinib in the absence or presence of AMG 176 (1 μM) and viability was determined after 3 days. Each dot represents an independent biological replicate ($N = 3$, ****$p = 0.00001$, unpaired-parametric $t$ test, 2-sided). **H, I** Cells were transfected with the indicated siRNAs and then treated with trametinib (0.1 μM) or sotorasib (1 μM) for 48 h, AMG 176 for 4 h or trametinib (0.1 μM) or sotorasib (1 μM) for 48 h followed by AMG 176 for 4 h. **J** siPP1B restores sensitivity (ΔAUC) to combined sotorasib or trametinib + AMG 176. Each dot represents an independent biological replicate ($N = 3$, ****$p = 0.00001$, 2-way ANOVA). **K** HA-tagged WT NUAK1 (WT) or GKKK NUAK were over-expressed in H2030 isogenic cells and the interaction of NUAK1 and PP1B was assessed by Co-IP. Source data are provided as a Source Data file.

increased intracellular AMP/ATP ratio and phosphorylating AMPK, which in turn suppresses energy consumption by inhibiting mTOR and stimulating autophagy[45]. Recently, the AMPK-related SIK kinases have been shown to play a major role in mediating the suppressive effects of LKB1 on tumorigenesis and metastatic potential in models of *KRAS*-mutant NSCLC[28,29]. However, a role for LKB1 in regulating apoptotic priming is largely undefined. To identify the LKB1 substrate kinase(s) that mediate the suppressive effect of LKB1 on drug-induced JNK

activation and MCL-1 dependency, we simultaneously silenced the expression of multiple members within each AMPK-related kinase family that are expressed in NSCLC[29] (Fig. 3F, S7A–D). Silencing NUAK1 + 2 was sufficient to restore the sensitivity of H2030 cells to combined sotorasib or trametinib + AMG 176 to a similar level as LKB1-deficient H2030 cells (Fig. 3G, S7E). In contrast, silencing SIKs, AMPKs, or MARKs in the context of LKB1 re-expression did not restore drug sensitivity (Fig. 3G, S7F). Additionally, the difference in drug sensitivity

between LKB1-deficient and LKB1-restored cells was similar when cells were cultured in high or low/absent glucose conditions (Fig. S7G), consistent with a nutrient-independent mechanism. We validated that knock-down of NUAKs, but not the other LKB1 substrates, restored drug sensitivity in H23 and MGH1112-1 cells expressing LKB1 (Fig. S7H). Knockdown of NUAK1/2 restored drug-induced JNK phosphorylation in H2030, H23, MGH1112-1 cells expressing LKB1 to a similar level as their matched isogenic controls (Fig. 3H, S8A, B), and increased the apoptotic response of LKB1-expressing cells to trametinib + AMG 176 (Fig. S8C).

NUAKs regulate cell polarity[46], ploidy[47], and adhesion[48] through phosphorylation of the myosin phosphatase targeting-1-protein phosphatase-1beta (PP1B) complex. NUAK1 directly binds to and activates the PP1B phosphatase by displacing the self-inhibitory protein I-2[48]. We hypothesized that PP1B activation downstream of LKB1-NUAK1 could lead to dephosphorylation of JNK. Knockdown of PP1B expression dramatically increased pJNK in LKB1-restored H2030 cells (Fig. 3I) and increased sensitivity to MAPK + MCL-1 inhibition (Fig. 3J, Fig. S8D), suggesting that PP1B de-phosphorylates JNK and reduces MCL-1 dependence downstream of LKB1. To demonstrate whether NUAK1 directly interacts with PP1B in LKB1-expressing *KRAS*-mutant NSCLC cells, we expressed HA-tagged NUAK1 in H2030, H23, MGH1112-1 EV, and LKB1 cells. Co-immunoprecipitation (Co-IP) of PP1B revealed increased binding of NUAK to PP1B in H2030 LKB1 cells (Fig. 3K, compare lanes 1 and 3) that was disrupted by mutation of the NUAK GILK domain (GKKK) that has been previously demonstrated to mediate the NUAK-PP1B interaction[48] (Fig. 3K, compare lanes 3 and 5). Conversely, binding of the I2 protein to PP1B was diminished in H2030 LKB1 cells and increased in the presence of the NUAK GKKK mutant, consistent with LKB1-dependent competition between NUAK and I2 for binding PP1B. These results were recapitulated in H23 and MGH1112-1 cells, with increased interaction between NUAK1 and PP1B in LKB1 expressing cells compared to EV controls (Fig. S8E, F). Collectively, these results indicate that loss of LKB1-NUAK1/2 signaling leads to increased JNK signaling as a consequence of decreased PP1B phosphatase activity, resulting in increased sensitivity to combined MAPK + MCL-1 inhibition.

## JNK phosphorylates BCL-XL to drive an MCL-1 dependent state

Inhibition of MEK/ERK signaling leads to BIM accumulation and increases apoptotic priming in oncogene-driven cancers treated with various targeted therapies, driving cells into an MCL-1 and/or BCL-XL dependent state[15,17]. To confirm that LKB1 modulates apoptotic priming, we performed BH3 profiling[49–51] on isogenic LKB1-deficient or WT cell lines before and after treatment with trametinib (Fig. S9A). As expected, trametinib treatment increased overall apoptotic priming (Fig. S9B). Trametinib induced a greater increase in MCL-1 specific priming (expressed as "Δ priming") in LKB1-deficient compared to LKB1 wild-type cells, and which was consistently reduced upon re-expression of LKB1 (Fig. S9C, D). Conversely, deletion of LKB1 in H358 cells increased trametinib-induced MCL-1 dependency. In a subset of cell lines, we also observed changes in BCL-XL dependency, however this was not a consistent effect (Fig. S9E). To investigate the basis for increased MCL-1 dependent priming in LKB1-deficient cells, we examined MCL-1 protein expression levels, as this is highly dependent on cap-dependent translational regulated by mTOR[52] (which is regulated by AMPK). Consistent with an AMPK-independent effect of LKB1, MCL-1, and BCL-XL protein expression was similar in LKB1-deficient and wild-type *KRAS*-mutant NSCLC cell lines (Fig. S9F, G) or isogenic cell line pairs (for example, see Fig. S10E). Next, we examined interactions between BIM and MCL-1 or BCL-XL. Co-IP experiments revealed increased BIM bound to MCL-1 and BCL-XL after trametinib treatment (Fig. S10A–C), consistent with prior studies[16]. LKB1-deficient cells treated with trametinib had a greater amount of BIM bound to MCL-1, and less BIM bound to BCL-XL, compared to LKB1 wild-type cell lines

(Fig. S10A–C). Restoration of LKB1 in deficient cell lines reduced the amount of BIM bound to MCL-1 after trametinib treatment, and knocking out LKB1 in wild-type cells increased the amount of BIM bound to MCL-1 (Fig. 4A, S10D–G). Notably, except for one cell line (A427), the impact of LKB1 re-expression/knock-down on baseline BIM:MCL-1 binding was less prominent in the absence of drug treatment. These results indicate that loss of LKB1 promotes the formation of BIM:MCL-1 complexes, especially in the context of suppression of oncogenic MAPK signaling, functionally inducing an MCL-1 dependent state, and priming AMG 176 sensitivity.

MCL-1 and BCL-XL can be phosphorylated at multiple residues by numerous kinases, including JNK and ERK, leading to context-specific and divergent effects on protein stability/degradation, BIM binding affinity, and apoptosis[53–58]. MCL-1 phosphorylation at T163 decreased acutely upon trametinib treatment consistent with a loss of ERK phosphorylation[59] and then rebounded at later time points coinciding with activation of JNK (Fig. S11A). Restoration of LKB1 in LKB1-deficient cells reduced the rebound in MCL-1 phosphorylation, while deleting LKB1 in wild-type cells increased MCL-1 phosphorylation (Fig. S11A, B). A similar time and JNK-dependent pattern of phosphorylation of BCL-XL at S62 was observed in LKB1-deficient cells, which was suppressed by re-expression of LKB1. Upon treatment with the combination of trametinib + AMG 176, BCL-XL S62 was rapidly phosphorylated in LKB1-deficient but not LKB1-proficient isogenic cell line pairs (Fig. 4B). Silencing JNK1/2 expression reduced drug-induced phosphorylation of both MCL-1 and BCL-XL to a similar level as the corresponding LKB1-restored isogenic cell line (Fig. S11C, compare lanes 3, 4 and 7). To assess whether JNK-mediated phosphorylation of MCL-1 or BCL-XL impacts drug sensitivity, we expressed DOX-inducible MCL-1 or BCL-XL phosphorylation-site mutants in H2030 cells while simultaneously knocking down expression of endogenous MCL-1 or BCL-XL (Fig. 4C, S11D–G). While mutating MCL-1 phosphorylation sites to alanine had little effect on sensitivity to trametinib + AMG 176 (Fig. 4D, S11H), expression of the BCL-XL S62A mutant reduced sensitivity to both sotorasib or trametinib + AMG 176 in H2030 and other cell lines (Fig. 4E, F, S11K), phenocopying LKB1 re-expression and JNK1/2 knockdown. Conversely, the BCL-XL S62E phosphomimetic increased the sensitivity of H2030 LKB1 cells (Fig. 4G). These results suggest that the increased MCL-1 dependency of LKB1-deficient cells is mediated by BCL-XL phosphorylation.

## JNK phosphorylation alters BIM:BCL-XL interaction

Prior studies have demonstrated that sensitivity of cancer cells to MCL-1 inhibition is inversely related to BCL-XL expression level and the capacity for BCL-XL to neutralize pro-apoptotic BH3 proteins such as BIM[60,61]. Phosphorylation of BCL-XL S62 induces a conformational change in which a dysregulated domain folds into the BCL-XL BH3 binding groove to prevent BIM binding[58]. Therefore, we hypothesized that phosphorylation of BCL-XL S62 by JNK compromises the ability of BCL-XL to sequester BIM that is liberated from MCL-1 upon MCL-1 inhibition. To test this, we studied the dynamics of BIM:MCL-1 and BIM:BCL-XL interactions by first treating cells with trametinib to increase BIM bound to MCL-1, then treating with a short pulse of AMG 176 and assessing the ability for BCL-XL to sequester BIM released from MCL-1 (Fig. 5A). In LKB1-deficient H2030 cells, very little BIM was sequestered by BCL-XL upon treatment with AMG 176, compared to LKB1 wild-type SW1573 cells, which exhibited substantial sequestration of BIM by BCL-XL (Fig. 5B). Restoring LKB1 expression or silencing JNK1/2 in H2030 cells increased the amount of BIM sequestered by BCL-XL after addition of AMG 176 (Fig. 5C, D). In H2030 and MGH1112-1 EV cells, the BCL-XL S62A mutant exhibited increased BIM:BCL-XL binding, whereas in H2030 LKB1 cells, the phospho-mimetic S62E mutant decreased BIM:BCL-XL binding (Fig. 5E, F, S11L). Knock-down of NUAK1/2 expression in H2030 cells, which we showed restored drug-induced JNK phosphorylation (Fig. 3H), restored the drug-

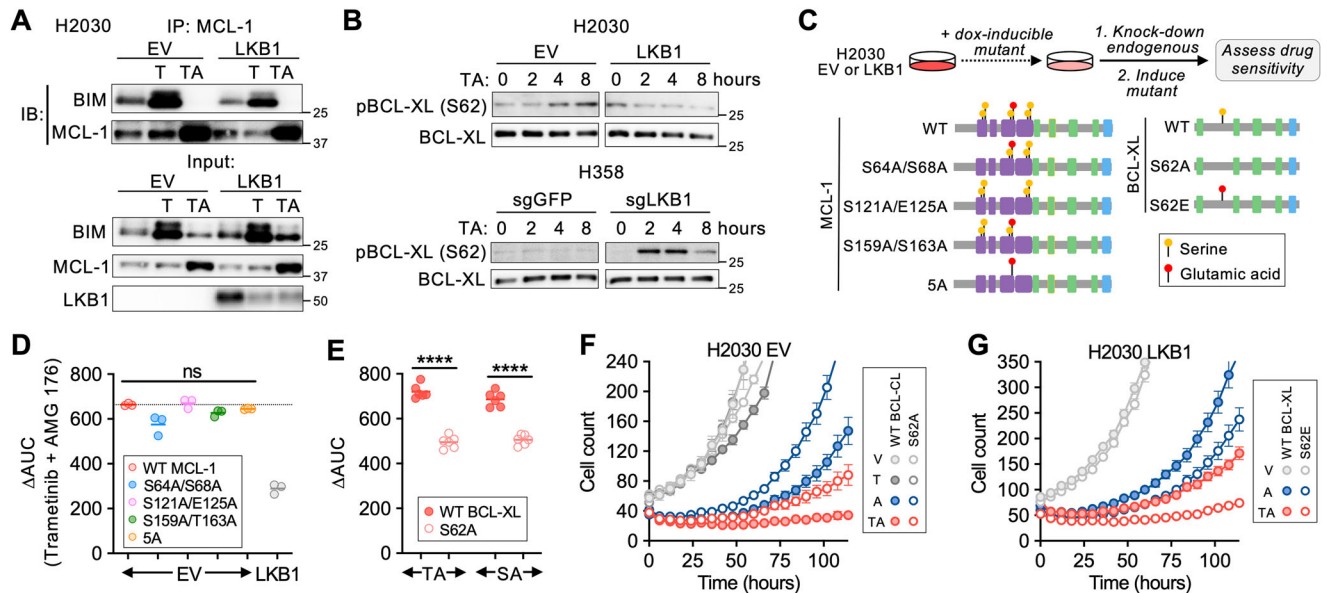

**Fig. 4 | JNK phosphorylates BCL-XL to drive an MCL-1 dependent state. A** Co-IP of BIM bound to MCL-1 in H2030 EV (empty vector) and H2030 LKB1 cells after treatment with vehicle, trametinib (0.1 μM) for 24 h or trametinib for 24 h followed by AMG 176 (1 μM) for 4 h. **B** Time course of BCL-XL S62 phosphorylation in isogenic H2030 and H358 cells by western blot after treatment with 0.1 μM trametinib + 1 μM AMG 176. **C** Experimental approach for expressing MCL-1 & BCL-XL phospho-site mutants while suppressing endogenous MCL-1 and BCL-XL. Interrogated phosphorylation sites are designated in yellow, phosphomimetic sites in red. **D** MCL-1 phospho-site mutants do not reduce sensitivity to MCL-1 inhibition (ΔAUC). After induction of mutant MCL-1 (or WT control) and knockdown of endogenous MCL-1, H2030 EV cells were treated with trametinib in the absence or presence of AMG 176 (1 μM) and viability was determined after 3 days. Each dot is an independent biological replicate (N = 3). **E** BCL-XL S62A mutant decreases MCL-1 dependence. After induction of BCL-XL S62A (or WT control) and knockdown of endogenous BCL-XL, H2030 EV cells were treated with sotorasib or trametinib alone or in the presence of AMG 176 (1 μM) and viability was determined after 3 days. Each dot is an independent biological replicate (N = 6, ****p = 0.000001, unpaired-parametric t test, two-sided). H2030 EV cells expressing inducible WT or S62A mutant BCL-XL S62A (**F**) or H2030 LKB1 cells expressing inducible WT or BCL-XL S62E phosphomimetic (**G**) were treated with 0.1 μM trametinib or 0.1 μM trametinib in combination with 1 μM AMG 176 and cell number was quantified by live-cell imaging. Data are mean and S.E.M. of 3 technical replicates. V vehicle, T trametinib, A AMG 176, TA trametinib + AMG 176. Source data are provided as a Source Data file. Western blots and immunoprecipitation images are representative of at least 2 independent biological replicates.

induced phosphorylation of BCL-XL S62 (Fig. 5G). Collectively, these results demonstrate that in the context of LKB1 loss, activation of JNK creates an MCL-1 dependent state by phosphorylating BCL-XL and decreasing its capacity to buffer the pro-apoptotic effects of BIM (Fig. 5H). While in some cases, especially those that may be highly primed and MCL-1 dependent at baseline, LKB1 loss may confer sensitivity to MCL-1 inhibition alone, MCL-1 dependency is enhanced by the increase in apoptotic priming upon suppression of oncogenic MAPK signaling.

## LKB1-deficient patient and PDX tumors are MCL-1 dependent

To investigate the clinical relevance of our findings, we performed BH3 profiling on *KRAS*-mutant NSCLCs (solid metastatic lesions or tumor cells isolated from malignant pleural effusions of patients) after ex vivo exposure to sotorasib or trametinib (Fig. 6A). Both sotorasib and trametinib treatment led to an increase in MCL-1 dependent priming (MS1 peptide) in *STK11/LKB1*-mutant but not WT tumors, (Fig. 6B, S12A). Consistent with this effect, co-immunoprecipitation experiments performed on tumor cells isolated from a malignant pleural effusion obtained from the same patient revealed drug-induced increases in BIM bound to MCL-1 (Fig. 6C). In contrast, we did not observe a significant difference in drug-induced BCL-XL dependent priming (HRK peptide) between *STK11*-mutant and WT tumors. To extend these findings, we performed BH3 profiling on *KRAS*-mutant (G12C and other) NSCLC patient-derived xenograft (PDX) models with or without co-occurring *STK11* loss after short-term treatment with trametinib. Similar to the patient tumors and in vitro cell line models, LKB1-deficient tumors exhibited increased MCL-1-dependent priming compared to WT tumors (Fig. 6D, S12B). The addition of AMG 176 to

sotorasib led to greater tumor response than sotorasib alone in LKB1-deficient PDX tumors with MCL-1-dependent priming but not LKB1-deficient PDX tumors (Fig. 6E, F, S12C, D). LKB1-deficient PDX tumors exhibited pronounced JNK phosphorylation in response to sotorasib + AMG176 treatment, which was not observed in LKB1 wild-type PDX tumors (Fig. S13A). We confirmed a similar increase in phospho-JNK in drug-treated LKB1-deficient, but not LKB1-expressing, isogenic H2030 xenograft tumors (Fig. S13B). To investigate potential toxicity, we assessed a combination dosing regimen with intermittent AMG 176 administration (AMG 176 is administered as intermittent infusions in currently on-going clinical trials) that induced similar tumor regression (Fig. S13C). In humanized MCL-1 knock-in mice[61] the combination of sotorasib with AMG 176 was well tolerated with no overt signs of toxicity (Fig. S13D). Consistent with the expected effects of on-target MCL-1 inhibition[61], we observed decreased B cells and monocytes, however no additional effects were observed in combination with sotorasib compared with AMG 176 alone (Fig. S13E). Thus, loss of the LKB1 tumor suppressor is associated with increased MCL-1 dependence upon treatment with sotorasib or trametinib in *KRAS^G12C*-mutant NSCLCs, creating an apoptotic vulnerability that can be exploited by concurrent inhibition of MCL-1.

## Discussion

While the utility of targeting truncal oncogenic driver mutations in lung cancer is firmly established, most clinical targeted therapy strategies do not take into account co-occurring mutations. For *KRAS*-mutant lung cancers in particular, identifying vulnerabilities associated with recurring co-occurring mutations in tumor suppressor genes could enable the development of biomarker-driven combination therapies with enhanced activity in distinct subsets of patients.

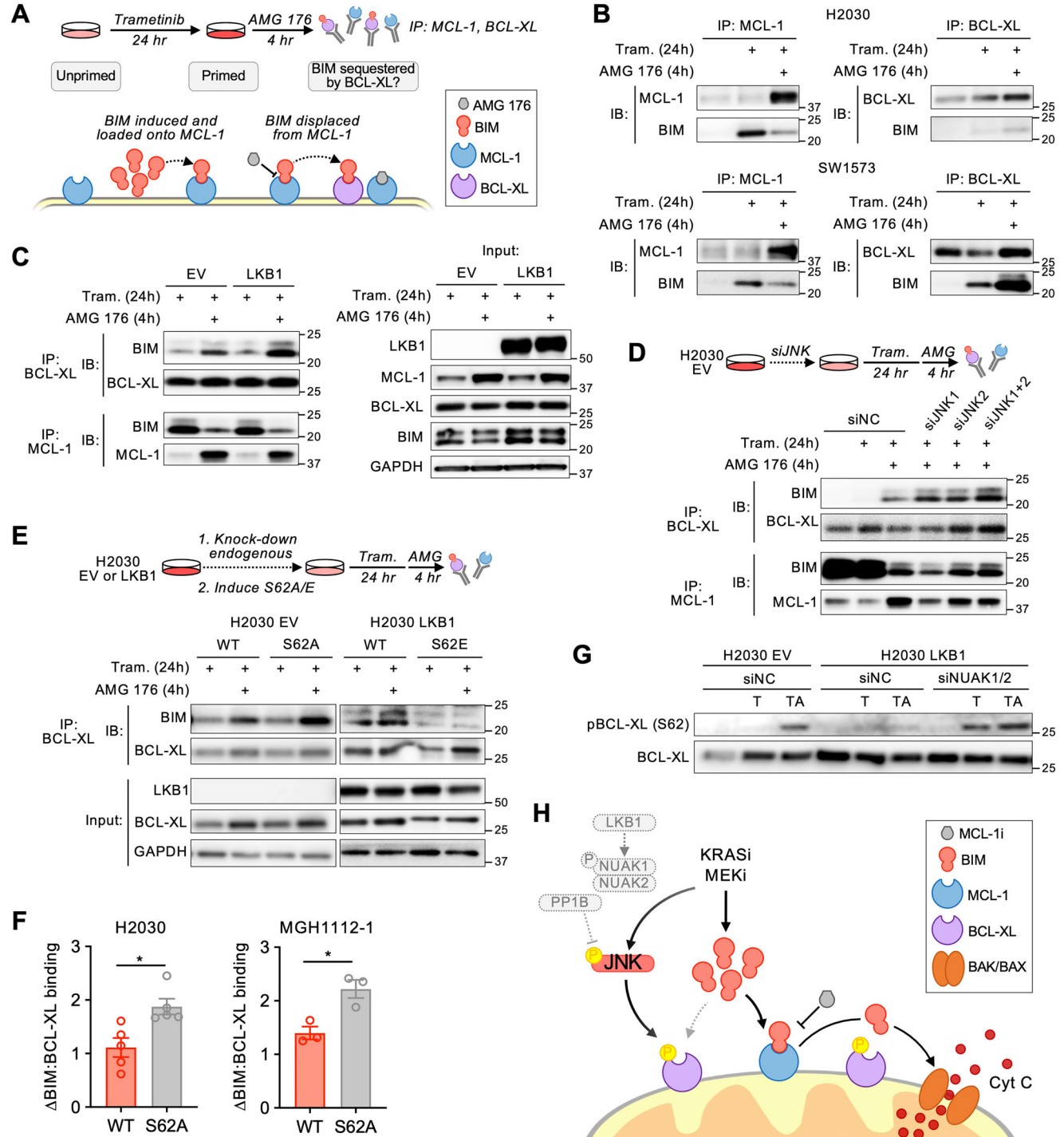

**Fig. 5 | JNK activation drives an MCL-1 dependent state by modulating BIM:BCL-XL interactions. A** Scheme for approach to investigating BIM sequestration upon displacement from MCL-1. **B** Co-IP assessment of BIM bound to MCL-1 and BCL-XL in H2030 (LKB1-deficient) and SW1573 (LKB1 wild-type) cells after treatment with 0.1 μM trametinib for 24 h followed by 1 μM AMG 176 for 4 h. **C** Co-IP assessment of BIM bound to BCL-XL and MCL-1 in H2030 EV and LKB1 cells after treatment with 0.1 μM trametinib for 24 h followed by 1 μM AMG 176 for 4 h. **D** Co-IP assessment of BIM bound to BCL-XL and MCL-1 in H2030 EV (empty vector) with JNK knockdown after treatment with 0.1 μM trametinib for 24 h + 1 μM AMG 176 for 4 h. **E** Co-IP assessment of BIM bound to WT BCL-XL or BCL-XL mutants in H2030 EV (S62A) and H2030 LKB1 (S62E) cells after treatment with 0.1 μM trametinib for 24 h followed 1 μM AMG 176 for 4 h. HA-tag pull downs are specific for inducible

constructs. **F** Quantification of Co-IP assessment of BIM bound to BCL-XL in H2030 and MGH1112 cells overexpressing BCL-XL WT or S62A mutants. Data are mean and S.E.M., each dot represents a biological replicate ($N = 3$-5, H2030: *$p = 0.0433$, MGH1112-1: *$p = 0.0345$, unpaired-parametric $t$ test, 2-sided). **G** Effect of NUAK1/2 knockdown on BCL-XL S62 phosphorylation in response to treatment with 0.1 μM trametinib for 48 h (T) or trametinib for 48 h followed by 1 μM AMG 176 (TA) for 4 h. **H** Model depicting the mechanism by which LKB1 loss leads to an MCL-1-dependent state and sensitizes *KRAS*-mutant NSCLCs to combined KRAS or MEK + MCL-1 inhibition. Source data are provided as a Source Data file. Western blots and immunoprecipitation images are representative of at least 2 independent biological replicates.

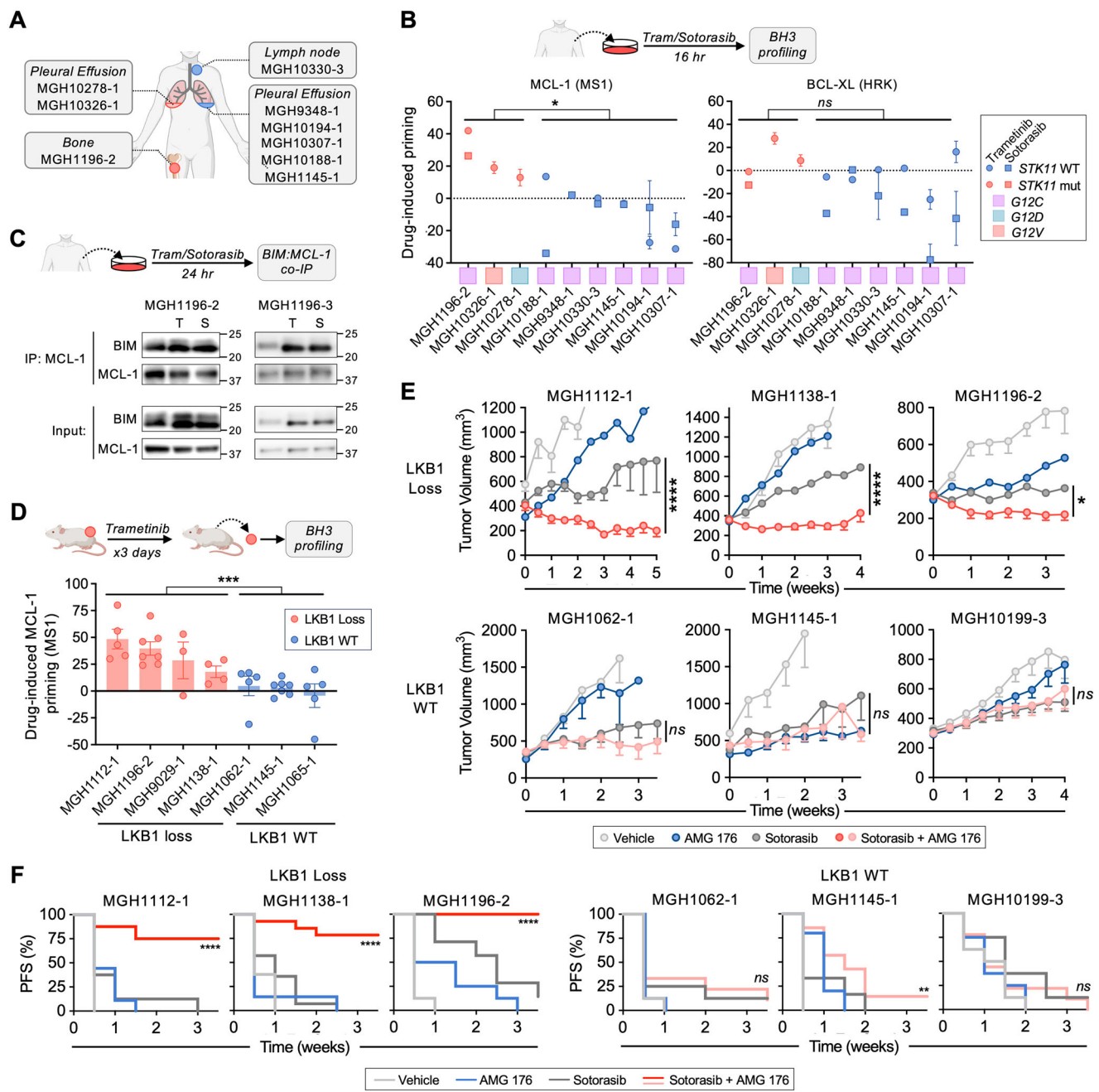

**Fig. 6 | LKB1 loss in associated with MCL-dependence of *KRAS^G12C*-mutant NSCLC PDX tumors and patient tumor explants. A** *KRAS^G12C*-mutant NSCLC tumor cells were collected for BH3 profiling and assessment of BIM:MCL-1 interactions after ex vivo treatment with sotorasib or trametinib. **B** Change in MCL-1 (MS1 10 + 30 μM peptide) and BCL-XL (HRK 10 + 100 μM peptide) dependent priming of patient tumor cells after ex vivo treatment with 0.1 μM trametinib or 1 μM sotorasib treatment. Data is normalized to Vehicle control and error bar presents S.E.M. (*N* = 3,6. *\*p* = 0.0476, unpaired-nonparametric *t* test, 2-sided). **C** Co-IP assessment of BIM:MCL-1 interaction in tumor cells isolated from pleural fluid after ex vivo treatment with 0.1 μM trametinib (T) or 1 μM sotorasib (S) for 16 h. Data is representatives of 2 independent biological replicates. **D** Mice bearing *KRAS^G12C*-mutant NSCLC patient derived xenograft (PDX) tumors were treated with sotorasib (100 mg/kg) for 3 days and harvested for BH3 profiling. Data shown is the difference in MCL-1

dependent priming (MS1 peptide) of sotorasib treated tumors normalized to vehicle control, each dot represents an independent mouse tumor with error bar as S.E.M. (*N* = 3–7, *\*\*\*p* = 0.0002, 2-way ANOVA). **E** Mice bearing *KRAS^G12C*-mutant NSCLC PDX tumors (LKB1-loss: MGH1112-1, MGH1138-1, MGH1196-2; LKB1 WT: MGH1062-1, MGH1145-1, MGH10199-3) were treated with vehicle, sotorasib (100 mg/kg), AMG 176 (50 mg/kg) or sotorasib (100 mg/kg) + AMG 176 (50 mg/kg) daily. Data shown are mean and S.E.M. of *N* = 6–10 animals per arm, statistical difference between single agent and combination arms was determined using mixed-effects model, *\*\*\*\*p* = 0.00001, *\*p* = 0.013, 2-way ANOVA). **F** Survival curves of PDX tumor-bearing mice: the progression free survival (PFS) metric was determined by time to 20% increase in tumor volume from baseline measurement (*\*\*\*\*p* = 0.0001, *\*\*p* = 0.0087, Log-rank Mantel-Cox test). Source data are provided as a Source Data file. **A–D** Created in BioRender. Li, C. (2025) https://BioRender.com/p50l332.

---

However, the development of the most KRAS inhibitor drug combinations currently in the clinic has been agnostic to co-occurring mutations. Our finding that LKB1 regulates the apoptotic dependency of *KRAS*-mutant lung cancers is unexpected, as genomic features associated with sensitivity to BH3 mimetics in oncogene-addicted solid

tumors have been elusive[12,16,17]. Inactivating mutations or loss of *STK11*/LKB1, which define one of the major genomic sub-groups of *KRAS*-mutant lung cancers[1,19,62], are of particular interest because they are associated with decreased responsiveness to immune checkpoint blockade[9,63] and poor overall prognosis[64].

LKB1 is a master kinase that regulates diverse cellular processes via phosphorylation of multiple members of AMPK family kinases[45,65]. In particular, the role of LKB1 in regulating energy homeostasis via AMPK has been well defined. In settings of energy stress (high AMP:ATP ratio), AMPK limits anabolic processes by inhibiting mTORC1 through TSC2[66]. Interestingly, expression levels of MCL-1 are highly dependent upon mTOR-mediated cap-dependent translation, and inhibition of mTOR by small-molecule inhibitors has been shown to reduce MCL-1 expression and confer apoptotic sensitivity[33]. We also observed an association between PI3K inhibition, MCL-1 down-regulation, and AMG 176 sensitivity in LKB1-deficient *KRAS*-mutant NSCLC cell lines. However, we did not observe any change in MCL-1 expression upon manipulation of LKB1, and silencing AMPK expression did not phenocopy the effect of LKB1 loss on MCL-1 inhibitor sensitivity. Collectively, these results support an AMPK-independent mechanism by which LKB1 modulates JNK signaling and MCL-1 dependency.

Beyond its role regulating metabolism via AMPK, LKB1 loss promotes tumorigenesis by reprogramming epigenetic states, facilitating lineage plasticity and promoting metastasis[22,67–70]. Recent studies have revealed a central role for the AMPK-related SIK kinases in mediating the suppressive effects of LKB1 on tumorigenesis, with potential contributing role for NUAKs[28,29]. NUAK kinases have been shown to regulate cellular polarity, adhesion, and cell cycle in normal tissues[46,48,71] and to play a critical role in neurite formation[72]. Our results reveal that NUAKs can function as negative regulators of JNK signaling, through binding and activation of the JNK phosphatase PP1B. To our knowledge, the LKB1/NUAK1/PP1B axis represents a mechanism by which LKB1 can suppress JNK stress signaling and regulate apoptosis. JNK has been reported to modulate apoptotic signaling by phosphorylating multiple pro- and anti-apoptotic BCL-2 family members, including BIM[73–76], BAX[77–79], BCL-XL[56,57] and MCL-1[53,55,80,81]. The consequences of differential phosphorylation are complex and can impact both protein stability/turnover as well as protein-protein interactions, leading to both pro- and anti-apoptotic effects in a context-specific manner. We observed JNK-mediated phosphorylation of both MCL-1 and BCL-XL in response to KRAS and MEK inhibition, however elimination of JNK phosphorylation sites in BCL-XL but not MCL-1 phenocopied the decrease in MCL-1 dependence observed with JNK knockdown or LKB1 re-expression. Future studies will be necessary to determine whether JNK phosphorylation of MCL-1 may confer apoptotic vulnerabilities in other therapeutic contexts. Interestingly, we observed that a subset of LKB1-deficient cell lines exhibited sensitivity to single agent MCL-1 inhibition in the absence of MAPK inhibition, indicative of a highly-primed MCL-1-dependent baseline state. Re-expression of LKB1 partially decreased sensitivity to MCL-1 inhibition, suggesting that the baseline suppression of JNK by LKB1/NUAK may impact apoptotic dependency in the absence of therapeutic stress in some cases, which is further amplified by the increased apoptotic priming that occurs in the setting of suppression of oncogenic MAPK signaling.

While our study focused on *KRAS*-mutant lung cancers treated with KRAS or MEK inhibitor targeted therapies, we also provide evidence that LKB1 suppresses JNK activation in response to UV radiation, suggesting a fundamental role for LKB1 in regulating JNK stress signaling in response to a variety of stimuli. From an evolutionary perspective, we speculate that the ability for LKB1 to suppress JNK signaling may be advantageous in normal tissues facing energy or redox stress by temporarily suppressing apoptosis until compensatory mechanisms (also regulated by LKB1) can be engaged. It is less clear whether modulation of JNK signaling contributes to the tumor suppressor functions of LKB1, or whether the ability to hyperactivate JNK signaling provides an advantage to cancer cells with loss of LKB1. It is notable that the differential JNK activation and increase in MCL-1 dependency conferred by LKB1 loss was maximally observed in the setting of MAPK inhibition, suggesting that the functional effects of

this pathway may be unmasked in specific contexts in response to select perturbations. Furthermore, we observed that knock-down of LKB1 suppressed drug-induced JNK activation to varying degrees in different cell lines, suggesting that additional context-specific pathways may cooperate to regulate JNK.

In summary, we identify a mechanism by which LKB1-NUAK regulates JNK stress signaling and modulates apoptotic dependencies in *KRAS*-mutant NSCLCs. In response to KRAS or MEK inhibition, LKB1-deficient cells exhibit hyperactivation of JNK and increased reliance on MCL-1 to buffer the increase in BIM. While LKB1-deficiency does not confer increased sensitivity to KRAS[G12C] or MEK inhibitors used as single agents, cells become primed for apoptosis when treated with MCL-1 BH3 mimetics. These results suggest a potential biomarker-informed combination therapy approach based on mutations or genomic loss of *STK11*/LKB1.

## Methods

### Cell culture

Patient-derived NSCLC cell lines were established in our laboratory from malignant pleural effusions, core-needle biopsies or surgical resections using methods that have been previously described (PMID: 30254092), except for the MGH1070 cell line that was derived from a primary mouse PDX model. All patients provided informed consent to participate in a Dana-Farber/Harvard Cancer Center Institutional Review Board–approved tissue collection protocol and granted permission for research to be conducted on their samples. Clinically observed *KRAS* mutations (determined by MGH SNaPshot NGS genotyping panel) were verified in established cell lines. Established patient-derived cell lines were maintained in RPMI + 10% FBS. Publicly available NSCLC cell lines were obtained from the Center for Molecular Therapeutics at the Massachusetts General Hospital Cancer Center; STR validation was performed at the initiation of the project (Biosynthesis, Inc.). Cell lines were routinely tested for mycoplasma during experimental use. Cell lines were maintained in RPMI + 5% FBS except A427, SW1573, H2009 and H1573 which were maintained in DMEM/F12 + 5% FBS.

### Cell proliferation assessment

Cell proliferation was assessed using the CellTiter-Glo assay (Promega). Cells were seeded into 96-well plates 24 h prior to drug addition. Cell proliferation was determined 72 h after addition of drug by incubating cells with CellTiter-Glo reagent for 30 min at room temperature on a shaking platform. Luminescence was quantified using a SpectraMax i3x plate reader (Molecular Devices).

### PI/Annexin apoptosis assay

Cells were seeded at low density 24 h prior to drug addition. Seventy-two hours after adding drugs, adherent (alive) and floating (dead) cells were collected, stained with propidium iodide (PI) and Cy5-Annexin V (BD Biosciences) and analyzed by flow cytometry. The annexin-positive apoptotic cell fraction was quantified using FlowJo software.

### Generation of engineered cell lines

**EV and LKB1 cell lines.** Empty vector (EV, pBabe) and LKB1 retro-viral vectors were gifts from Dr. Kwok-Kin Wong (NYU). EV and LKB1 virus were prepared by transfecting HEK293T cells with EV or LKB1, VSV-G (Addgene #8454), Gag-Pol (Addgene #14887) using Lipofectamine 3000 (ThermoFisher) and collecting viral particles in the supernatant. Stable cell lines were generated by transducing *KRAS*-mutant NSCLC lines with EV or LKB1 virus followed by puromycin selection.

**LKB1 knock-out cell lines.** sgRNAs targeting the *STK11* locus were designed using CHOP-CHOP and cloned into pSpCaS11(BB)-2A-GFP (Addgene #48138). *KRAS*-mutant NSCLC cell lines were transiently transfected with the plasmids and sorted for single clone formation

by FACs. After clonal expansion, 20 clones were selected and loss of LKB1 expression was assessed by western blot. Alternatively, LKB1 sgRNAs were cloned into lentiCRISPR v2 (Addgene #52961). Lentiviral particles were prepared by transfecting HEK293 cells with EV or sgLKB1, VSV-G (Addgene #8454) and Δ8.91 using Lipofectamine 3000 (ThermoFisher). Stable cell lines were generated by infecting *KRAS*-mutant NSCLC lines with lentiCRISPR v2 or sgLKB1 virus followed by puromycin selection.

**DOX-inducible MCL-1, BCL-XL cell lines.** Full length wild-type or mutant MCL-1, BCL-XL coding sequences were synthesized (GenScript) and cloned into pInducer20 (gift from Lee Zou, MGH). Lentiviral particles were prepared by transfecting HEK293 cells with pInducer20 or pInducer20-MCL-1/ pInducer20-BCL-XL, VSV-G (Addgene #8454) and Δ8.91 using Lipofectamine 3000 (ThermoFisher). Stable cell lines were generated by infecting *KRAS*-mutant NSCLC lines were infected with EV or pInducer20-MCL-1 or pInducer20-BCL-XL virus followed by selection with neomycin/G418.

### Mouse xenograft studies

*KRAS-mutant* NSCLC PDX models were generated by subcutaneous implantation of tumor cells or tissue from malignant pleural effusions, core-needle biopsies or surgical resections into male NSG mice (Jackson Labs). All patients signed informed consent to participate in a Dana Farber/Harvard Cancer Center Institutional Review Board-approved protocol giving permission for research to be performed on their sample. All animal studies were conducted through MGH Institutional Animal Care and Use Committee (IACUC)–approved animal protocols in accordance with institutional guidelines. Subcutaneous tumors were serially passaged twice to fully establish each model. Clinically observed *KRAS* mutations were verified in each established model. All mice used were between 8 weeks and 6 months of age. Mice were kept at a temperature of $74 \pm 2\,°F$ ($23 \pm 1\,°C$) with a 12-h light/12-h dark cycle. Relative humidity was maintained at 30–70%. Mice numbers used in each experiment are indicated in the figure legends. Maximum tumor size did not exceed $2000\,mm^3$, in accordance with IACUC regulations. For drug studies, PDX tumor tissue was directly implanted subcutaneously into NSG or athymic nude (Nu/Nu). For H2030 xenograft studies, cell line suspensions were prepared in 1:1 matrigel:PBS, and $5 \times 10^6$ cells were injected unilaterally into the subcutaneous space on the flank of female nude (Nu/Nu) mice. Tumors were measured with electronic calipers, and the tumor volume was calculated according to the formula $V = 0.52 \times L \times W^2$. Mice with established tumors (250–400 mm$^3$) were randomized to drug treatment groups using covariate adaptive randomization to minimize differences in baseline tumor volumes. Trametinib was dissolved in 0.5% HPMC/0.2% Tween 80 (pH 8.0) and administered by oral gavage daily at 3 mg/kg, 6 days per week. Sotorasib was dissolved in 2% HPMC/0.1% Tween 80 (pH 7) and administered by oral gavage daily at 100 mg/kg, 6 days per week. AMG 176 was dissolved in 25% hydroxypropylbeta- cyclodextrin (pH8.0) and administered by oral gavage daily 50 mg/kg.

### Quantitative RT-PCR analysis

RNA was extracted using the Qiagen RNeasy kit. cDNA was prepared with the Transcriptor High Fidelity cDNA Synthesis Kit (Roche) using oligo-dT primers. Quantitative PCR was performed with gene specific primers (Supplementary Data 2) using SYBR™ Select Master Mix (Applied Biosystems) on a Lightcycler 480 (ThermoFisher). Relative gene expression was calculated by using the Δ ΔCT method by normalizing to *ACTB*.

### Western blot analysis

Cells were seeded in either 6-well or 6 cm plates and drug was added when cells reached 70% confluency. Cells were harvested by washing twice with PBS, lysed in lysis buffer[16] on ice, and spinning at 14,000

RPM at 4 °C for 10 min to remove insoluble cell debris. Lysate protein concentrations were determined by a bicinchoninic acid assay (ThermoFisher). Gel electrophoresis was performed using NuPage 4–12% Bis-Tris Midi gels (Invitrogen) in NuPage MOPS SDS Running Buffer (Invitrogen) followed by transfer onto PVDF membranes (Thermo-Fisher). Following transfer, membranes blocked with 5% milk (Lab Scientific bioKEMIX) in Tris Buffered saline with Tween 20 (TBS-T) and then incubated with primary antibody (1:1000, 1%BSA in TBS-T) at 4 °C overnight. After washing in TBS-T, membranes were incubated with the appropriate secondary antibody (1:12500 in 2% skim milk in TBS-T) for 1h at room temperature. The following HRP-linked secondary antibodies were used: anti-rabbit IgG (CST7074) and anti-mouse IgG (CST7076). Membranes were removed from secondary antibodies and washed 3 times for 10 min each in TBS-T. Prior to imaging, membranes were incubated for 4 min SuperSignal West Femto Stable Peroxide & Luminol/Enhancer (Thermo Fisher) diluted 1:10 in 0.1 M tris-HCL pH 8.8 (Boston Bioproducts). Luminescence was imaged using a G:Box Chemi-XRQ system (Syngene). The following primary antibodies were used: pJNK T183/Y185 (CST4668), SAPK/JNK (CST9252), BIM (CST2933), pBCL-XL S112 (Invitrogen 44-428 G), BCL-XL (CST2764), LKB1 (CST3050), pMCL-1 T163 (CST14765), pMCL-1 S159/T163 (CST4579), pMCL-1 S114 (CST13297), MCL-1 (BD Pharmingen 559027), pMKK4 S257/T261 (CST9156), MKK4 (CST9152), pMEK7 S271 (Thermo Fisher PA5-114604), pMEK7 T275 (ThermoFisher PA5-114605), MKK7 (CST4172), DUSP10/MKP5 (CST3483), HA Tag (CST3724), β-Tubulin (CST2146), GAPDH (CST5174). All antibodies for Western blot analysis were diluted to a concentration of 1:1000. Figure 3H: The samples derive from the same experiment but different gels for LKB1, pJNK, and JNK. The samples derive from the same experiment but different gels for PP1B and GAPDH. Figure 3K: The samples derive from the same experiment but different gels for PP1B and I-2. Figure 5B–E: The samples derive from the same experiment but different gels for BIM and BCL-XL. Fig. S5B: The samples derive from the same experiment but different gels for LKB1 and pAMPK. Fig. S6A, D, H: The samples derive from the same experiment but different gels for LKB1, pJNK, JNK and GAPDH. Fig. S8A, B: The samples derive from the same experiment but different gels for pJNK, JNK and GAPDH. Fig. S8E, F: The samples derive from the same experiment but different gels for PP1B and GAPDH. Fig. S9F: The samples derive from the same experiment but different gels for LKB1, pAMPK and ACTIN. Fig. S10 A, B, E: The samples derive from the same experiment but different gels for BIM and BCL-XL. Fig. S11A, B: The samples derive from the same experiment but different gels for pMCL-1, pJNK and GAPDH. Fig. S11C: The samples derive from the same experiment but different gels for LKB1, pJNK, pMCL-1 and GAPDH. Fig. S11F: The samples derive from the same experiment but different gels for BCL-XL and HA. Fig. S11G: The samples derive from the same experiment but different gels for pMCL-1 S64, pMCL-1 S159 and pMCL-1 T163. Fig. S11L: The samples derive from the same experiment but different gels for BIM and BCL-XL.

### Protein Immunoprecipitation

Cells were seeded in either 10 cm or 15 cm plates and drug was added when cells reached 70% confluency. Cells were harvested after the treatment period and lysates were prepared using tris lysis buffer with Protease Inhibitor Cocktail (Meso Scale Discovery) on ice. After normalization of total protein concentrations, Pierce Protein A/G Magnetic Beads (ThermoFisher) and either mouse anti-human MCL-1 (4 µg/reaction, BD Pharmingen 559027) or mouse anti-human BCL-XL (4 µg/reaction, EMD Millipore MAB3121) antibodies were added to lysate aliquots and incubated at 4 °C overnight. A representative aliquot of the normalized whole cell lysate was saved for western blot analysis. The immunoprecipitated fractions were separate using magnetic separation, washed three times with tris lysis buffer on ice, proteins eluted by heating at 95 °C for 10 min with tris lysis buffer and LDS sample buffer 4X (Invitrogen).

For western blots, the rabbit anti-human MCL-1 (1:1000, CST4572) antibody was used; all other antibodies were identical to those used for western blotting. For immunoprecipitation of HA-tagged BCL-XL, the Pierce Magnetic HA-Tag IP/Co-IP Kit (Thermo Fisher) was used following the manufacturer's protocol (specifically, the procedure for (A.) Manual IP/Co-IP and (B.) Elution Protocol 2 for reducing gel analysis).

## Immunofluorescence and image analysis

Cells were fixed with 10% neutral-buffered formalin and permeabilized by PBST (PBS + Triton X100). Cells were then incubated with pJNK T183/Y185 (CST4668) primary antibody (1:200) overnight at 4°C. Secondary antibody staining was performed at room temperature for 1 h, followed by DAPI staining. Images were acquired using a Zeiss LSM 710 confocal microscope. Image analysis was performed using CellProfiler software (Broad Institute). Briefly, individual cells were identified by DAPI staining. pJNK staining inside the nuclei or outside the nuclei was segmented and quantified at the individual cell level.

## Immunohistochemistry

Xenograft tumors were dissected and fixed in 4% paraformaldehyde (Sigma 158127). Paraffin embedding and sectioning were performed by the MGH Histopathology Core. Tissue slides were baked at 60 °C for 10 min followed by dehydration. Antigen retrieval was performed by incubating in citrate-based solution (vector lab H-3300-250) in a pressure cooker. Slides were then incubated in 3% hydrogen peroxide in methanol for 10 min and washed twice with distilled water. Slides were then incubated in blocking solution (pJNK and pAMPK: 3% BSA in TBS-T, LKB1: Cas-block Thermofisher 8120) followed by primary antibody diluted 1:200 in TBS-T (pJNK CST 4668S, pAMPK CST 2535S, LKB1 CST 3050S) overnight at 4 °C. Following primary antibody incubation, slides were washed with TBS-T and incubated with secondary biotinylated antibody and VECTASTAIN Elite ABC Reagent (vector lab PK-6101). HRP detection was performed using DAB Substrate Kit (vector lab SK-4100) and counterstained with hematoxylin (Sigma 51275). Slides were then dehydrated and mounted. Tissue imaging was performed using a Leica DM2500 microscope. TBS-T: Tris-buffered saline with Tween.

## siRNA-mediated gene knockdown

siRNA transfection was performed using Lipofectamine RNAiMAX Transfection Reagent according to the manufacturer's protocol (Invitrogen, Cat# 13778075). In brief, cells were seeded in 6-well, 6 cm, or 10 cm plates and siRNA transfection was carried out when cells reached ~70% confluency. Prior to transfection, cells were placed in antibiotic-free media. 48 h after transfection, cells were seeded for analysis of proliferation or immunoprecipitation or harvested for western blot. The following Invitrogen siRNA were used: Negative Control (NC) (AM4611), MAPK8 (ID: s11152), MAPK9 (ID:s11159), NUAK1 (ID: S110), NUAK2 (ID: s37779), PRKAA1 (ID: S120), PRKAA2 (ID: s11056), PRKAB1 (ID: s11059), PRKAB2 (ID: s11062), SIK1 (ID: S115377), SIK2 (ID: s23355), SIK3 (ID: s23712), MARK1 (ID: S12511), MARK2 (ID: S11648), MARK3 (ID: S12514), MARK4 (ID: s33718), MAP2K4 (ID: s11182, s11183), MAP2K7 (ID: s11183, s11184), MCL-1 (ID: S12584, S12585), BCL2L1 (ID: s1920, s1921, s1922).

## BH3 profiling of cell lines

BH3 profiling was performed by quantifying cytochrome c release upon addition of exogenous BH3 peptides as previously described[49]. Briefly, $2 \times 10^6$ cells were isolated, centrifuged at $500 \times g$ for 5 min, then the cell pellet was resuspended in 100 μL PBS with 1 μL Zombie Green viability dye (BioLegend, cat# 423111). Cells were stained at room temperature out of light for 15 min, then quenched by addition of 400 μL FACS Stain Buffer (2% FBS in PBS). Cells were then centrifuged

at $500 \times g$ for 5 min and subjected to BH3 profiling with indicated peptides and concentrations. After BH3 profiling, cells were permeabilized for intra-cellular staining with a saponin-based buffer (1% saponin, 10% BSA in PBS) and stained with an antibody for cytochrome c AlexaFluor 647 (BioLegend 612310; 1:2000 dilution) and DAPI overnight at 4 °C. Cells were analyzed flow cytometry (Attune NxT) the following day.

## BH3 profiling of primary patient samples

All patients signed informed consent to participate in a Dana Farber/Harvard Cancer Center Institutional Review Board-approved protocol giving permission for research to be performed on their sample. Patients were between the ages 44–78 with 6 (67%) female and 3 (33%) male. Surgical resection tumor tissue was manually minced to ~1 mm³ and incubated in RPMI1640 + 10% FBS overnight in the absence or presence of drugs. Immediately prior to BH3 profiling, tissue was further dissociated by collagenase/dispase enzymatic dissociation for 30 min at 37 °C. Samples were then strained through a 100 μM filter to isolate single cells. For each sample, $2 \times 10^6$ cells were isolated, centrifuged at $500 \times g$ for 5 min, and the cell pellet was resuspended in 100 μL PBS with 1 μL Zombie Green viability dye (BioLegend, cat# 423111). Cells were stained at room temperature out of light for 15 min, then quenched by addition of 400 μL FACS Stain Buffer (2% FBS in PBS). Cells were centrifuged at $500 \times g$ for 5 min and resuspended in 100 μL FACS Stain Buffer. Cells were then stained with the following conjugated cell-surface marker antibodies at 1:50 dilutions: EpCAM PE (BioLegend, 324206) and CD45 BV786 (BioLegend, 304048). Cells were then centrifuged at $500 \times g$ for 5 min and subjected to BH3 Profiling as previously described[49] with indicated peptides (e.g., MS1 = MCL-1, HRK = BCL-XL) and concentrations. After BH3 profiling, cells were permeabilized for intra-cellular staining with a saponin-based buffer (1% saponin, 10% BSA in PBS) and stained with an antibody for cytochrome c AlexaFluor 647 (BioLegend, 612310; 1:2000 dilution) and DAPI. Cells were stained overnight at 4 °C and analyzed by flow cytometry (Attune NxT) the following day. Cells of interest were identified: DAPI positive, Zombie negative, CD45 negative, and EpCAM positive.

## Phosphoproteomic analysis

All phospho-proteomic samples were analyzed as single biological replicates. Cell pellets frozen, lysed, reduced with DTT followed by alkylation with iodoacetamide. Proteins were precipitated following the MeOH/CHCl3 protocol, digested with trypsin and LysC. Phopho-peptide enrichment was performed as described previously[39,82,83]. Phosphopeptide enrichment was performed with TiO2 beads (GL Sciences, Japan) from 2.5 mg of peptides for each sample. Phospho-peptides labeling was performed with TMT10plex reagents (Thermo Fisher Scientific), then samples were pooled and fractionated by basic pH reversed phase chromatography into 24 fractions as previously described[84]. Fractionated peptides were dried, dissolved in 5% ACN/5% formic acid, and analyzed via 3-h LC-M2/MS3 runs on an Orbitrap FusionLumos mass spectrometer using the Simultaneous Precursor Selection (SPS) supported MS3 method[85,86] as previously described[87]. For each peptide, two MS2 spectra were acquired using CID and HCD fragmentation[88]. Each MS2 spectra was assigned using a SEQUEST-based custom proteomics analytical pipeline[89] allowing phosphorylation of tyrosine, threonine, and serine residues as a variable modification. Correct assignment of phosphorylation within a peptide sequence was evaluated using the Ascore algorithm[90]. Protein and peptide assignments were filtered to a false discovery rate (FDR) of <1% using the target-decoy database search strategy[91] and employing linear discriminant analysis and posterior error histogram sorting[89]. Peptides sequences corresponding to more than one protein in the UniProt database (2014) were assigned to the protein with the greatest number of matching peptides[89]. For MS3 spectra, only those with an average

signal-to-noise value > 40 per reporter ion and an isolation specificity[86] > 0.75 were quantified. Protein TMT-intensities were normalized by a two-step process. P rotein intensities were first normalized over all TMT channels for each protein according to the median average protein intensity for all proteins. Slight mixing errors from each sample in peptide mixture were corrected by calculating the median of the normalized intensities from all proteins in each TMT channel, then individual protein intensities were normalized to the median intensity values.

## Proteomic analysis

Peptides from the tryptic digest described above (50 μg) were labeled using TMT-10plex reagents (Thermo Scientific). After pooling, labeled samples were fractionated by basic reversed phase HPLC[84]. Fractions were analyzed via 3 h reversed phase LC-MS2/MS3 runs on an Orbitrap FusionLumos. Protein identification from from MS2 spectra was performed using the SEQUEST algorithm searching across a human data base (uniprot 2014)[92] using a custom analytical pipeline[89]. The search strategy included a target-decoy database-based search to filter against a false-discovery rate (FDR) of <1%[91]. MS3 isolation for quantification used Simultaneous Precursor Selection (SPS) as described previously[85–87]. Only MS3 with a signal-to-noise value of >40 per reporter ion and an isolation specificity[86] of >0.75 were considered; normalization using a two-step normalization process was performed as described above.

## Phosphoproteomic signature analysis

Phospho-signature analysis was performed using PTM-Signature Enrichment Analysis (PMT-SEA), a modified version of ssGSEA2.0 (https://github.com/broadinstitute/ssGSEA2.0). Briefly, relative log-fold increases/decreases were calculated by comparing the levels of phospho-peptides in each group. Relative log-fold increases/decreases were imported into the PMT-SEA package and compared against the PTM signatures database (PTMsigDB). Significant signatures were exported, ranked and compared between groups (for example LKB1-positive versus LKB1-negative isogenic pair).

## Synergy analysis

Synergy analysis was performed using Biochemically Intuitive Generalized Loewe (BIGL)[93]. NSCLC cell lines were treated according to a 2 × 2 matrix of increasing doses of trametinib/sotorasib and AMG 176 for 72 h and cell viability was assessed by CellTiter-Glo. Synergy analysis was divided performed three steps: 1. Marginal curve was determined for each compound using non-linear least squares estimation. 2. The expected effect for "General Loewe model" was computed from previously computed marginal curve. 3. The expected response was compared with observed viability using maxR statistical test, which evaluates whether the null model locally fits the observed data.

## Statistics and reproducibility

Statistical testing for all experiments was performed using student *t* test, one-way or two-way ANOVA. For student *t* test, either paired, unpaired, parametric, or parametric testing was performed for individual experiments as specified in the corresponding figure legends. All tests were performed using two-sided hypothesis. Two-way ANOVA was used for multiple comparisons of groups with Tukey correction (95% confidence interval). Data shown for western blots and immunoprecipitation experiments is representative of at least two independent biological replicates. Images of immunofluorescence and immunohistochemistry experiments are representative of at least 3 biological replicates. No statistical method was used to predetermine sample size. No data were excluded from the analyses. Investigators were not blinded to allocation during experiments and outcome assessment.

## Reporting summary

Further information on research design is available in the Nature Portfolio Reporting Summary linked to this article.

## Data availability

Raw phosphoproteomic data generated in this study have been deposited in MassIVE [https://massive.ucsd.edu/ProteoSAFe/static/massive.jsp] under accession code "MSV000097246", as well as Proteomexchange [https://www.proteomexchange.org/] under accession code "PXD061550, doi:10.25345/C5HH6CJ48". Processed phophoproteomic data (normalized intensity) can be downloaded from Harvard Dataverse using identifier "https://doi.org/10.7910/DVN/OLVIT7". All other raw data, including graphs and western blots, are provided in the Supplementary Information/Source Data file. All flow cytometry gating strategies are provided in the Supplementary Information/FACs Gating file. Source data are provided with this paper.

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

## Acknowledgements

We thank the patients and their families who provided tumor tissue for analysis and generation of patient-derived models. We thank members of the Hata lab and MGH Thoracic Oncology Group for helpful discussion and support. We thank Dr. Kwok-Kin Wong for providing the pBABE-EV and pBABE-LKB1 plasmids. We thank Dr. Nabeel Bardeesy for project guidance and providing the CRTC2/CREB target gene oligos. This study was funded by support from the NIH F32 CA250231 (C.L.), Mark Foundation for Cancer Research (The EXTOL Project) (A.N.H.), Stand Up To Cancer–American Cancer Society Lung Cancer Dream Team Translational Research Grant (SU2C-AACR-DT17-15) (A.N.H.), the Ludwig Center at Harvard (A.N.H.), a research grant provided by Amgen, Inc. (A.N.H.), and by Be a Piece of the Solution Research Fund at MGH. ASCO Conquer Cancer Foundation Young Investigator Award (C.S.N); AACR 20-40-15-NABE (C.S.N); Career Enhancement Award from Dana Farber/Harvard Cancer SPORE in Lung Cancer (P50CA265826) (C.S.N); Charles W. and Jennifer C. Johnson Clinical Investigator Program at the Koch Insitute of Massachusetts Institute of Technology (C.S.N).

## Author contributions

C.L, M.U.S., A.N, M.D.V, Y.S., C.F., Z.I., X.Q. J.O., J.K., S.E.C, G.K., E.H., R.L., A.O., M.W., performed the experiments. R.H., J.L. contributed human samples and/or data. S.E.C., G.K., E.H., A.O., M.W. developed patient-derived cell lines and PDX models. C.L., R.M. performed data analysis and interpretation. S.C., A.Y.S., K.R., J.R.L., P.E.H. provided KRAS and MCL-1 inhibitors used in this study. C.L., J.O., C.N., K.S., W.H., P.E.H., A.N.H. were involved with study design. C.L. wrote the manuscript. C.L. and A.N.H. edited the manuscript. All authors discussed the results and commented on the manuscript.

## Competing interests

A.N.H. has received research support from Amgen, Blueprint Medicines, BridgeBio, Bristol-Myers Squibb, C4 Therapeutics, Eli Lilly, Novartis, Nuvalent, Pfizer, Roche/Genentech, Scorpion Therapeutics, and Triana Biomedicines; has served as a compensated consultant for Amgen, Engine Biosciences, Nuvalent, Oncovalent, Pfizer TigaTx, and Tolremo Therapeutics. K.S. received research funding from Gate Bioscience and Dialectic Therapeutics. RSH has served as a compensated consultant for Abbvie, Amgen, Astrazeneca, Biohaven, Claim, Daichii Sankyo, EMD Serono, Gilead, Lilly, Merck, Novartis, Regeneron, Sanofi. Research funding to institution, not to self: Abbvie, Agios, Corvus, Daichii Sankyo, Exelixis, Genentech, Lilly, Mirati, Novartis, Turning Point. JJL has served as a compensated consultant for Genentech, C4 Therapeutics, Blueprint Medicines, Nuvalent, Bayer, Elevation Oncology, Novartis, Mirati Therapeutics, AnHeart Therapeutics, Takeda, CLaiM Therapeutics, Ellipses, AstraZeneca, Bristol Myers Squibb, Daiichi Sankyo, Yuhan, Merus, Regeneron, Pfizer, Roche, Gilead, Janssen, Nuvation Bio, Eli Lilly, AstraZeneca, Gilead, and Turning Point Therapeutics; has received institutional research funds from Hengrui Therapeutics, Turning Point Therapeutics, Neon Therapeutics, Relay Therapeutics, Bayer, Elevation Oncology, Roche, Linnaeus Therapeutics, Nuvalent, and Novartis; and travel support from Pfizer, Merus, Takeda, and Bristol Myers Squibb. C.S.N. owns equity (stock) in Opko Therapeutics and has received royalty income from Cambridge Epigenetix. S.C., A.Y.S., K.R., R.L., B.B., J.R.L., P.E.H. are employees of and have ownership (including stock, patents, etc.) interest in Amgen. A.Y.S. also owns stock from Abbvie. The remaining authors declare no competing interests.
