## [Transparent Peer Review file · Nature Communications]

LKB1 regulates JNK-dependent stress signaling and apoptotic dependency of KRAS-mutant lung cancers

Corresponding Author: Dr Aaron Hata

Version 0:

Reviewer comments:

Reviewer #5

(Remarks to the Author)

I have been asked to review this manuscript, although I did not review it in the first round. Thus, as the stand-in Reviewer #3 I have tried not to come up with an entirely new review but rather focus on the response to original Reviewer #3's comments.

I totally agree with Reviewer #3's original comments. The manuscript should have never even been submitted with key data from only one cell lines and without consistent experiments throughout all figures. The authors have largely addressed this point now and have added lots of new data.

The original minor comment about the colors used in Figure 1 is still validate. I find the labelling very annoying and really, it's not hard to just label like everyone else does. Maybe you are worried that the figure will get crowded with all that labelling, then just split Figure 1 into two figures. As is, I spent too much time trying to figure out what was what if it was light or dark blue, red, or was gray.

Small things that should be fixed

Figure 1G, did you really KO GFP? Or is just sgGFP versus sgLkb1?

Line 10 perhaps new Heymach Nature paper should be added here as a ref.

Lines 19/20, 28/29 and many other places. They state that the effects are independent from co-occurring mutations, but they need to qualify this statement that their screen has a very limited number of cell lines and thus very poor resolution to find effect.

Lines 301/302, that comment is not correct. Data from that pair of CD papers documented the role of Nuaks in tumor suppression. Perhaps they actually mean something else, in which case they should change that sentence

REVIEWERS' COMMENTS

Reviewer #5 (Remarks to the Author):

I have been asked to review this manuscript, although I did not review it in the first round. Thus, as the stand-in Reviewer #3 I have tried not to come up with an entirely new review but rather focus on the response to original Reviewer #3's comments.

I totally agree with Reviewer #3's original comments. The manuscript should have never even been submitted with key data from only one cell lines and without consistent experiments throughout all figures. The authors have largely addressed this point now and have added lots of new data.

The original minor comment about the colors used in Figure 1 is still validate. I find the labelling very annoying and really, it's not hard to just label like everyone else does. Maybe you are worried that the figure will get crowded with all that labelling, then just split Figure 1 into two figures. As is, I spent too much time trying to figure out what was what if it was light or dark blue, red, or was gray.

We thank the reviewer for their suggestions regarding the formatting of Figure 1. We have now split it into new Figures 1 and 2, as suggested. Additionally, we have separated some of the overlapping data in Figure 1F as well as simplified the color scheme improved clarity and more diverse colors. Furthermore, we have separated data in the prior Figure 1J (now new Figure 2D) for improved clarity and labeling.

Small things that should be fixed

Figure 1G, did you really KO GFP? Or is just sgGFP versus sgLkb1?

We have changed all the KO GFP and KO LKB1s to sgGFP and sgLKB1, respectively across all figures and text.

Line 10 perhaps new Heymach Nature paper should be added here as a ref.

We have now added this reference (Reference #10, Line 13)

Lines 19/20, 28/29 and many other places. They state that the effects are independent from co-occurring mutations, but they need to qualify this statement that their screen has a very limited number of cell lines and thus very poor resolution to find effect.

We have added the qualifying statements "in our limited cell line cohort", "did not appear to impact sensitivity" and similar wording in each of these instances each to avoid overstating the role of co-occurring mutations in cell line sensitivities. (Lines 64, 68, 74).

Lines 301/302, that comment is not correct. Data from that pair of CD papers documented the role of Nuaks in tumor suppression. Perhaps they actually mean something else, in which case they should change that sentence.

We have modified the sentences to reflect a potential contributing role for NUAKs in tumorigenesis. (Line 371)